# A Systematic Review on Autism and Hyperserotonemia: State-of-the-Art, Limitations, and Future Directions

**DOI:** 10.3390/brainsci14050481

**Published:** 2024-05-10

**Authors:** Dario Esposito, Gianluca Cruciani, Laura Zaccaro, Emanuele Di Carlo, Grazia Fernanda Spitoni, Filippo Manti, Claudia Carducci, Elena Fiori, Vincenzo Leuzzi, Tiziana Pascucci

**Affiliations:** 1Department of Human Neuroscience, Unit of Child Neurology and Psychiatry, Sapienza University of Rome, Via dei Sabelli 108, 00185 Rome, Italy; dario.esposito@uniroma1.it (D.E.); filippo.manti@uniroma1.it (F.M.); 2Department of Dynamic and Clinical Psychology, and Health Studies, Sapienza University of Rome, Via degli Apuli 1, 00185 Rome, Italy; gianluca.cruciani@uniroma1.it (G.C.); grazia.spitoni@uniroma1.it (G.F.S.); 3Department of Psychology, Sapienza University, Via dei Marsi 78, 00185 Rome, Italy; laurazaccaro2@gmail.com (L.Z.); tiziana.pascucci@uniroma1.it (T.P.); 4Department of Experimental Medicine, Sapienza University, Viale del Policlinico 155, 00161 Rome, Italy; emanuele_dicarlo@yahoo.com (E.D.C.); claudia.carducci@uniroma1.it (C.C.); 5Cognitive and Motor Rehabilitation and Neuroimaging Unit, IRCCS Fondazione Santa Lucia, Via Ardeatina 306-354, 00179 Rome, Italy; 6Rome Technopole Foundation, P.le Aldo Moro, 5, 00185 Rome, Italy; elena.fiori@uniroma1.it; 7Centro “Daniel Bovet”, Sapienza University, P.le Aldo Moro 5, 00185 Rome, Italy; 8Fondazione Santa Lucia Istituto di Ricovero e Cura a Carattere Scientifico, Via Ardeatina 306, 00179 Rome, Italy

**Keywords:** autism spectrum disorder, hyperserotonemia, endophenotype

## Abstract

Hyperserotonemia is one of the most studied endophenotypes in autism spectrum disorder (ASD), but there are still no unequivocal results about its causes or biological and behavioral outcomes. This systematic review summarizes the studies investigating the relationship between blood serotonin (5-HT) levels and ASD, comparing diagnostic tools, analytical methods, and clinical outcomes. A literature search on peripheral 5-HT levels and ASD was conducted. In total, 1104 publications were screened, of which 113 entered the present systematic review. Of these, 59 articles reported hyperserotonemia in subjects with ASD, and 26 presented correlations between 5-HT levels and ASD-core clinical outcomes. The 5-HT levels are increased in about half, and correlations between hyperserotonemia and clinical outcomes are detected in a quarter of the studies. The present research highlights a large amount of heterogeneity in this field, ranging from the characterization of ASD and control groups to diagnostic and clinical assessments, from blood sampling procedures to analytical methods, allowing us to delineate critical topics for future studies.

## 1. Introduction

Autism spectrum disorder (ASD) is a complex and heterogeneous group of neurodevelopmental conditions characterized by persistent deficits in social communication and interaction, and restrictive and repetitive behavior patterns [1,2].

Approximately 1/36 8-year-old children in the US are estimated to be diagnosed with ASD [3], while the prevalence globally is about 1 per 100 children [4]. However, despite the prevalence and extensive research, considerable uncertainty regarding the exact pathophysiology of ASD and its impact on clinics remains [5,6,7]. Except a small percentage of individuals with syndromic (or secondary) ASD due to a specific identifiable cause (usually genetic, metabolic, or infectious) [8,9], approximately 85% of cases are considered idiopathic [9,10], namely without a known cause and most likely caused by complex gene-environment interactions rather than single major etiological factors.

This heterogeneity leads to different clinical presentations, severities, comorbidities, developmental outcomes, and difficulties in accurate diagnosis [7,11,12,13]. Even if diagnoses are rarely established before 12–18 months of life, generally symptoms appear in children younger than 2 years [2,14]. Among the various perspectives used to analyze this complex condition and to support early diagnosis and intervention, there is the identification of physiological biomarkers: considering ASD diffusion and diverse presentations, it is crucial to try to understand what is common for the target population in terms of biomarker levels [15,16,17,18]. The term “biomarker” has been used to define any indicator of normal biological or pathogenic processes or responses to exposure or treatment [19]. 

Hyperserotonemia, an elevated level of serotonin (5-hydroxytryptamine or 5-HT) in the blood, was the first biomarker to be discovered in autism research [20,21], and to date, this is one of the quantitative traits most consistently associated with the disorder in a significant subset of ASD patients [15,17,21,22,23,24]. Hyperserotonemia has been better described as an endophenotype of autism, being a measurable, familial, and heritable biomarker consistently associated with a complex disorder [6,25,26]. Endophenotypes can also be described as intermediate phenotypes that may be related to the underlying genetic components of a disorder but do not necessarily cause it directly [25,27].

Several hypotheses have been proposed to explain hyperserotonemia in autism. Physiologically, 5-HT is produced by enterochromaffin cells from the amino acid tryptophan in the gastrointestinal tract (GI) [28,29]. Once synthesized, 5-HT is released into the bloodstream, and most of it is stored in platelets via the 5-HT transporter (SERT) [30,31]. One possible mechanism of hyperserotonemia in patients with ASD is the increased density of SERT found in patients’ platelet membrane, leading to an increased uptake of 5-HT from the blood [17,31,32]. Dysregulation of the serotonergic system might affect immunological signaling cascades (e.g., cytokine-mediated regulation of SERT transcription and activity), impacting neural–immune interactions and leading to abnormalities in neuronal connectivity [33]. Several genes (e.g., SLC6A4 and ITGB3) expressing proteins regulating 5-HT metabolism both in the brain and platelets have been linked to autistic phenotypes [34,35,36,37], supporting the role of 5-HT in the pathogenesis of ASD. Furthermore, 5-HT plays a fundamental role as a trophic factor during prenatal neurodevelopmental processes [31,38], such that in utero exposure to drugs that increase blood levels of 5-HT (such as selective serotonin uptake inhibitors, valproic acid, alcohol, and cocaine) has resulted in higher rates of defective embryogenesis and ASD [33,39,40].

The systematic literature review and meta-analysis by Gabriele et al. [15] confirmed that elevated 5-HT blood levels are the most reliable and frequently replicated biomarker for ASD, regardless of the biomaterial studied (whole blood or platelet-rich plasma), whether 5-HT is normalized with the platelet count or not, and recommended the use of chromatographic techniques to measure 5-HT in whole blood normalized by the platelet count as the protocol.

Given the intense experimental activity and several new publications on the topic over the last decade, in the present work, a novel systematic review was performed to provide an up-to-date picture of the current research on hyperserotonemia and autism, as well as to systematically review the diagnostic approaches, laboratory methods, and clinical outcomes associated with elevated peripheral 5 HT in ASD. In particular, this systematic review was conducted with the primary aim of satisfying three needs: (1) verifying the consistency of hyperserotonemia as an endophenotype of ASD, (2) identifying the major clinical outcomes revealed by the subpopulation of hyperserotonemic ASD patients, and (3) providing recommendations to remove experimental obstacles and to help advance the research on this topic.

## 2. Materials and Methods

Details of the protocol for this systematic review were registered on PROSPERO (unique ID: CRD42023489735) and can be retrieved at www.crd.york.ac.uk/PROSPERO/ (accessed on 6 May 2024). The present review used PRISMA guidelines for systematic reviews and meta-analyses [41]. Figure 1 depicts the PRISMA flow diagram, describing the selection process in detail.

The study-retrieval and -selection process of eligible articles to be included in the current work followed three steps: First, a predefined algorithm was defined and used to search for suitable publications in scientific databases of interest. Second, duplicates were removed, and the titles and abstracts of the selected article were first screened: papers that passed screening procedure were deeply analyzed and assessed according to predefined inclusion and exclusion criteria. Third, data of interest were extracted from the selected articles. 

### 2.1. Database Search Strategy

A database search was conducted on the major scientific electronic databases in the field of health and social sciences: PubMed (https://pubmed.ncbi.nlm.nih.gov/ (accessed on 28 September 2023)), Scopus (https://www.scopus.com/search/ (accessed on 28 September 2023)), and PsycInfo (http://www.apa.org/pubs/databases/psycinfo/ (accessed on 28 September 2023)). Databases were searched for publications beginning with the inception of databases and up until 28 September 2023. The database search was conducted according to the following search algorithm: ((Autism OR autistic OR ASD OR “pervasive developmental disorder”) AND (hyperserotonemia OR serotonin OR 5-HT OR 5-hydroxytryptamine) AND (levels OR peripheral OR serum OR plasma OR blood OR platelets)). 

### 2.2. Literature Search Strategy and Study Eligibility

After article retrieval through the search algorithm, all duplicates were removed, and all articles were considered only once. The remaining publications underwent title and abstract screening to exclude articles that were not relevant to the topic. Subsequently, the remaining papers were checked according to predefined inclusion and exclusion criteria. Inclusion criteria were (a) studies with an analytical study design as defined by Grimes and Schulz [42] (e.g., an observational study with a comparison or control group) or providing comparisons with normative data—both retrospective and prospective studies were included to take into consideration the highest number of studies; (b) diagnosis of idiopathic ASD according to international diagnostic classification manuals; and (c) evaluation of 5-HT levels in the peripheral blood sample. Exclusion criteria were (a) studies conducted on samples of patients with ASD occurring in the context of a clinically and/or genetically defined syndrome (i.e., non-idiopathic ASD, e.g., fragile X syndrome or tuberous sclerosis complex); (b) studies written in other languages than English; (c) case reports, reviews, letters to the editor, commentaries, meeting abstracts, book chapters, dissertations, study protocols, and seminars; (d) animal models; (e) surgical protocols or validation of measurement instruments; and (f) for pharmacological and behavioral intervention trials, only pre-intervention baseline measures were considered. When the full text was not found, the article was excluded. The reference section of the included articles was checked to search for additional relevant literature; publications underwent the study eligibility process whenever appropriate citations were found. 

Two independent reviewers conducted the selection process. In the case of disagreement, they discussed their views until a consensus was reached. When necessary, an agreement was pursued involving a third reviewer.

### 2.3. Variables of Interest and Data Extraction

Articles that survived the selection process were then analyzed. Information about the following variables of interest was extracted: (a) authors and publication year, (b) ASD sample characteristics, (c) control sample characteristics, (d) diagnostic procedure, (e) biomaterial sampling procedures, (f) evaluated outcomes (i.e., measures of 5-HT levels; correlations between 5-HT levels and ASD-core symptoms; and neurocognitive, emotional–behavioral, or other relevant clinical outcomes in subjects with ASD), (g) study design, (h) significant findings, and (i) quality assessment. Quality assessment of the chosen publications was performed by employing a quality index derived and adapted from the Newcastle–Ottawa Scale (NOS) on a 9-star model. A full description of the modified NOS employed and on the quality assessment procedure is available in Appendix A [43].

## 3. Results

From the initial database search, 1150 potentially eligible articles were retrieved; subsequently, duplicates were removed, and a total of 811 manuscripts were screened by reading title and abstract. Six hundred eighty-one papers successfully passed the first step; inclusion and exclusion criteria were then applied for further screening of the articles. Of these, 116 fulfilled all the predefined criteria and were included in the present systematic review. Table 1 and Figure 2 show an overview of the general and demographic characteristics of the included manuscripts. Appendix A display further information about the quality assessment process.

### 3.1. Demographic Characteristics of the Studies

Sample sizes of ASD groups varied between *n* = 4 and *n* = 428, although information about sample size was not retrievable in one study. Globally, 7665 ASD patients from 24 countries were evaluated (most frequently from the US—47 studies; France—12 studies; Italy—7 studies; and The Netherlands and China—6 studies each). Eighteen studies did not report information about patients’ sex. Among the remaining studies, 5605 (82.00%) ASD patients were males, with 19 papers presenting data on male patients only. Six articles did not report information about patients’ age. Among the remaining, the mean age of ASD samples across the papers ranged between 3.93 ± 0.14 and 34.3 ± 7.6 years. 

Concerning the control groups, the sample sizes ranged between *n* = 4 and *n* = 959, for a total of *n* = 9516 controls evaluated. Of these, 4488 (47.16%) were unrelated, typically developed individuals; 3743 (39.33%) were ASD participants’ parents; 994 (10.45%) were ASD participants’ unaffected siblings; 24 (0.25%) were ASD participants’ unaffected relatives; and 267 (2.81%) were patients suffering from other psychiatric or neurological conditions. Control groups were age- and sex-matched in 41 (34.75%) studies. In 28 studies, data from ASD patients were not compared with any control groups. 

Quality indices of the included studies ranged between 1 and 9 points, with a median score of 6.

### 3.2. Diagnostic and Clinical Assessment

The detailed results are presented in Table 1 and Figure 3.

The diagnostic criteria used to diagnose ASD varied from study to study, clearly reflecting the wide range of years of the selected articles (1970–2022), with the DSM-IV and its text revision criteria being the most frequently used in 42 studies (Figure 3A). However, it should be noted that, since 2015, only 13 articles have explicitly adopted the DSM-5 criteria for ASD. Of all the selected articles, nine did not report the diagnostic criteria used for sample selection. Diagnoses were confirmed using a variety of diagnostic instruments (Figure 3B), with the ADI/ADI-R (Autism Diagnostic Interview) [154] and ADOS/ADOS-2 (Autism Diagnostic Observation Schedule) [155] being the most commonly used instruments. Notably, up to 35 studies did not report a specific diagnostic tool and relied solely on the assessment of clinician experts to diagnose ASD, while 18 studies required only the consensus of at least two specialists. As mentioned earlier, the focus of the present study was exclusively on idiopathic ASD; therefore, the methods to identify the non-idiopathic ASD patients in the included studies were also examined. Fifty-seven of them did not explicitly state how they identified and excluded subjects with secondary autism, while the other studies used a combination of clinical screening, routine laboratory tests, genetic testing, and neuroimaging. Regarding controls, only 12 of the studies reviewed used any type of test besides personal history or self-report to confirm that typically developing control subjects did not have autism spectrum disorder or autistic traits. 

### 3.3. Serotonin Analysis

Data the concerning 5-HT concentration in biological fluids were examined. In 95 (82%) studies, the results relating to the concentration of 5-HT were expressed quantitatively. In the remaining 18%, however, they were expressed only through graphs or qualitatively. Of the 95 studies that reported a quantitative result, 63 had a control population or normal reference values. In these papers, different units of measure were used to quantify the 5-HT concentration. In 68% of the studies, it was computed in ng/mL (multiples and submultiples); in 17%, in nmol/L; in 8%, in nmol/109 platelets (multiples and submultiples); in 1, µg/109 platelets; and in 1%, ng/mg of protein. In 5% of the retrieved papers, the unit of measurement was not expressed. Looking at the alteration of 5-HT values, 60 articles (52%) reported hyperserotonemia in the ASD population compared to healthy controls or reference values, 8 (7%) detected hyposerotonemia, 11 (9%) found no differences, and 5 (4%) reported contradictory results related to the different matrix examined, the control population used, and the different ways of normalizing the 5-HT value.

Finally, 32 (28%) studies did not report this information either due to the absence of a control population or because that was not the scope of the study (e.g., interventional study).

#### 3.3.1. Pre-Study Treatment, Sampling, and Pre-Analytical Procedures

Regarding pre-study drug treatment, only in 15 (12.9%) studies were ASD patients medication-free for more than one year at the time of blood sampling for 5-HT assessment. On the other hand, no data were reported in 40 papers (34.5%) for pre-study pharmacological treatments. In 24 (20.7%) studies, at least a subset of ASD patients received drugs that were not included in the experimental procedures. However, in 30 studies (25.8%), only specific classes of medication were considered (e.g., serotonergic medications or drugs with effects on the central nervous system). 

To describe the pre-analytical procedures, six categories were considered: (1) anticoagulant in the sampling tube (EDTA, heparin, sodium citrate, and dextrose citrate), (2) storage of the sample at low temperatures (−20 °C < temperature < −80 °C), (3) whether samples are processed in the dark, (4) use of antioxidants to preserve the sample (e.g., metabisulfite and sodium citrate), (5) fasting period before sampling, and (6) diet regimen. It was found that 45 (39%) papers reported the use of anticoagulants, most of all EDTA (81%); 42 (36%) studies stored the samples in the cold; 1 study processed the samples in the dark; and 8 used antioxidants. Fasting blood sampling was used in 24 (21%) studies; 18 (16%) articles controlled for the diet regimen, with 11 of them reporting specifically a 5-HT- and 5-Hydroxytryptophan (5-HTP)-free or low content diet. Twenty-three papers did not report any information about pre-analytical procedures, and 15 did not use any of the categories listed above. In 32 studies, more than one procedure was used. 

#### 3.3.2. Analytical Methods

In the included articles, different analytical methodologies were used to determine 5-HT. Of the 116 studies, more than 91% (106) reported reference to the method used or described it in detail. Most of the articles reported brief descriptions of the analytical procedure, but no one method of validation or verification of data. The methods used can be grouped into three broad categories: (1) chromatographic techniques, (2) spectroscopic techniques, and (3) immunoenzymatic techniques. Of these categories, the most represented is the chromatographic techniques, which were employed in 57% of the studies. The chromatographic methodologies included different procedures for extracting the analyte from the matrix (e.g., acid deproteinization or liquid extraction with organic solvents) and various detectors (e.g., electrochemical detector, mass spectrometers, or photodiode array detector). Of these, the most widely used analytical method was that described by Anderson et al. [156], employed in 35% of the papers in which a chromatographic methodology was used. The second most used analytical technique is spectroscopy, employed in 23% of the studies, particularly the method described by Yuwiler et al. [157]. Immunoenzymatic assays are the least represented (20%): within this group, 80% of the studies adopted a commercial kit, whereas the remaining used homemade reagents.

#### 3.3.3. Matrices

Four different matrices were used: (1) whole blood, (2) platelet-rich plasma (PRP), (3) platelet-poor plasma (PPP), and (4) serum. More than one matrix was used in 9% of the included papers. 

#### 3.3.4. Peripheral Serotonin Concentration

Figure 4 summarizes cumulative and matrix-related data emerging from the literature analysis. Appendix A shows more data regarding the studies analyzing 5-HT in different matrices.

#### 3.3.5. Statistical Analysis and Cut-Off Values of Hyperserotonemia

Several approaches have been used to define hyperserotonemia or the difference between patient and control populations. The normality of the distribution of 5-HT concentration values in the ASD population was studied in 100 out of 116 articles. Sixty-seven did not report details since it was computed to decide which statistical tests to use to compare populations. Moreover, 25 found a non-normal, and 8 found a normal distribution of this marker in the ASD population. In total, 69 out of 84 articles that reported results regarding differences in 5-HT concentration values used the significance of statistical tests to express hyperserotonemia, hyposerotonemia, or no difference. With reference to control values, eight articles defined hyperserotonemia if higher than at least 1.67 or 2 standard deviations of the mean, while six if exceeding the 95th percentile. Finally, based on literature or empirical data, 13 studies adopted a reference range from 122 to 325 ng/mL for normal 5-HT. Eighteen articles used more than one quantification of hyperserotonemia.

### 3.4. Serotonin Levels and Clinical Outcomes

Twenty-six articles presented results concerning correlations between 5-HT levels and ASD-core symptoms, neurocognitive, emotional–behavioral or other relevant clinical outcomes in subjects with ASD. Some papers studied more than one type of outcome. Table 2 synthesizes the main results.

#### 3.4.1. ASD Core Symptoms and Neurocognitive Outcomes

The first set of papers considered the alteration of 5-HT levels concerning ASD-core symptoms (i.e., impaired communication and social interaction; and restricted, repetitive, and stereotyped patterns of behavior). Eleven studies found a positive correlation between blood 5-HT levels and the higher global severity of ASD. All of them (except [59]) that rated symptoms with the CARS (Childhood Autism Rating Scale) are overall more recent than those who found no correlation, and they also have a higher average quality score (6.9) than those who found no correlation (6.4) or a negative correlation (5.7). Other (7) studies argue that hyperserotonemia may be correlated with high-functioning ASD, that it is not associated with intellectual disability, or even that it has no correlation with ASD global level of severity (9). However, there are also studies, for example, that by Ho and colleagues [137], which found a positive correlation between 5-HT levels and intellectual disability in ASD and a strong negative correlation in healthy controls.

Regarding specific cognitive functions, verbal IQ and memory appear to be negatively correlated with 5-HT levels [112,120]. Interestingly, Hranilovic et al. [92] found that ASD subjects with well-developed speech had 40% lower 5-HT levels than subjects with less developed or undeveloped speech. In addition, they also found positive correlations between high levels of 5-HT and lower verbal skills but not with the total CARS score and the overall severity profile. Hyperserotonemic patients also showed higher deficits in non-verbal communication (as measured by parental interviews ADI-R), delayed emergence of social smiling [82], and low social responsiveness (as measured by SRS) [77].

Hyperserotonemia is also associated with restricted and repetitive patterns of behaviors (motor or verbal stereotypies) and interests [24,72,82]. Furthermore, Sacco et al. [82] noted a negative correlation between 5-HT levels and neurodevelopmental delay and a positive correlation with stereotyped behaviors; however, the developmental quotient was not correlated with the levels of 5-HT in Hérault et al. [114]. In Chugani et al. [69], subjects with normal 5-HT levels had a greater chance of improving in “restricted and repetitive behaviors” ADOS scores than those with hyperserotonemia. Notably, Levin-Decanini et al. [78] described a correlation (although not significant) between the patients’ 5-HT levels and their parents’ Broad Autistic Phenotype scores (subscales of detachment, pragmatic language, and rigidity). A correlation was also found between the concentration of 5-HT and sensory abnormality scores [54,69,72,77,78,82,114,132]. 

#### 3.4.2. Emotional–Behavioral Outcomes

Self-aggression was the main emotional–behavioral outcome investigated in relation to blood serotonin in ASD. While three studies (with mean quality score of 3) found a significant inverse relationship between 5-HT levels and self-harm, one (with a quality index of 8) found a positive correlation between 5-HT levels and this clinical outcome. Cook et al. [120] found no correlation between self-injurious behaviors or decreased pain sensitivity and hyperserotonemia (article with a quality score of 7). Only one study found a significant association between 5-HT levels and hetero-aggressive behaviors [96], finding an inverse correlation with the Overt Aggression Scale (OAS) scores. 

Chakraborti et al. [52] found a gender difference in emotional expression when comparing scores for “adaptation to change” and “fear/nervousness”, with the score of the former being positively correlated to 5-HT levels in all subjects, while the latter only in females. In another study [114], patients with higher 5-HT levels were found to have higher levels of “lack of initiative”. Marler et al. [65] found higher compulsive behavior scores in the hyperserotonemic subgroup of ASD patients.

#### 3.4.3. Other Relevant Clinical Outcomes

Other variables associated with ASD and high 5-HT levels, such as susceptibility to sleep diseases, gastrointestinal disorders, and autoimmune diseases, have been studied. 

Regarding sleep problems, they were considered in two studies, although inversely correlated with serotonergic status. Hua et al. [54] showed a positive correlation between sleep disturbances (as measured by CSHQ) and 5-HT levels in patients with ASD. In Sacco et al. [82], there was a negative correlation between 5-HT levels and circadian dysfunction. In Kheirouri et al. [67], however, there was a correlation between sleep disturbances and the severity of core symptoms of autism, with 5-HT levels negatively influencing the latter. Furthermore, sleep-disordered breathing, bedtime stamina, sleep anxiety, parasomnias, daytime sleepiness, and total sleep scores were also related to high 5-HT levels. 

With regard to gastrointestinal problems, two authors reported a positive correlation between 5-HT levels and gastrointestinal disturbances. Bridgemohan et al. [16] found no statistically significant trend of higher platelet 5-HT in patients with gastrointestinal disturbances (constipation, diarrhea, and gastroesophageal reflux). Moreover, in Kheirouri et al. [67], no correlation was found between gastrointestinal disorders and ASD. These data were also confirmed by Marler et al. [65] and another recent study [47], claiming that 5-HT levels are not predictive of gastrointestinal disorders. 

As regards autoimmune diseases, the study of Mostafa et al. [80] found that patients with ASD have families with a history of autoimmune disorders in significantly higher percentages than controls, even if this parameter does not seem to be related to hyperserotonemia. However, the same authors previously hypothesized a correlation between autoimmune disorders and serotonergic status [80]. In addition, Sacco et al. [82] claimed that hyperserotonemia is significantly related to the presence of allergies in the family.

## 4. Discussion

In about half a century, autism has gone from a rare disorder to the most frequently studied and publicized neurodevelopmental disorder [4]. Individuals with ASD are characterized by core symptoms in the areas of communication and restricted/repetitive behaviors; however, everyone is very different from one other, thus suggesting that it is strategic to consider the different subpopulations of ASD patients. The request for early ASD treatment has grown in the last decades, but, to date, diagnosis is made solely through professionals’ behavior observation and not before the age of 18–24 months, generally after concerns expressed by caregivers, as there are no screening methods that are sufficiently sensible to us to identify children with ASD. For this reason, in the last decades, there has been an increase in the research on the biological markers of ASD. 

Hyperserotonemia has been extensively studied as a biological endophenotype of ASD, associated with the disease and partially heritable [22]. Despite the wealth of literature in this field, the causes and clinical outcomes of elevated 5-HT levels in individuals with ASD remain unclear. 

This systematic review provides an up-to-date overview of knowledge regarding 5-HT levels in individuals with ASD. The first issue to consider is the amplitude of the timeframe of the included studies, ranging from 1970 to 2022. From the outset, the authors had doubts about the reliability of 5-HT as a biological marker. Several studies were performed to confirm its role up to 1990; 30 articles were published in this period. The main aim in these years has been to understand if there was hyperserotonemia in autistic patients and if this could constitute a separate subgroup of patients. From 1990 to 2000, 20 articles were published in which the research of hyperserotonemia in autistic patients continues, with a greater interest in discovering the possible causes of this endophenotype. After this relative decrease in research efforts between 1990 and 2000, there was a new growth from 2000 until today. While it is not clear what prompted this research, it is undoubtedly in the latter period that research on this endophenotype’s behavioral or symptomatic characterization began. Finally, from 2000 to 2022, 65 articles were published in which the analysis of 5-HT, in addition to having become a parameter to control when researching the biological correlates of ASD, was aimed at understanding whether the presence or absence of behavioral or symptomatic outcomes is correlated with hyperserotonemia.

This systematic review was conducted with the following main aims: to confirm consistency of hyperserotonemia as an endophenotype of ASD, to identify clinical outcomes mainly observed in hyperserotonemic ASD patients, and to provide experimental recommendations for future research on this topic. Unlike previous reviews, pharmacological studies were also included when they included 5-HT assessment at baseline. Most of these studies were of high quality and controlled for several different variables, adding robustness to the review and allowing us to explore a larger sample size (116 articles that meet the inclusion criteria). 

Concerning the first point of the review objectives, the analysis of publications highlighted the need for a single and uniform definition of hyperserotonemia. Indeed, it was defined variably as the condition in which the 5-HT values are above a fixed cut-off, or the values exceed the 95th percentile, as well as the mean plus 1.67 or 2 standard deviations obtained from control population. The indication of a unique approach to defining hyperserotonemia is essential to improve the comparability and generalizability of study findings. Nevertheless, the results confirm a general association between high 5-HT levels and ASD. As previously described [15], most included articles demonstrate a significant increase in 5-HT levels in individuals with ASD, with only 18 out of 113 studies reporting no difference in 5-HT levels or hyposerotonemia compared to neurotypical individuals. Thus, about the first aim, the consistency of hyperserotonemia as an endophenotype of ASD can be confirmed. As shown in Figure 4, independently from matrices (four different matrices were used for 5-HT level evaluation—whole blood, PRP, PPP, and serum—although most articles used whole blood), hyperserotonemia was found in a large part of the autistic population (ranging from about 44% in PPP to 72% in PRP). The literature agrees that more than 95% of the body’s 5-HT is outside of the CNS [158], primarily stored within the platelet in dense granules [30], and the dependency of the number of platelets with the age, sex, and ethnicity could affect the 5-HT values. When establishing 5-HT reference values, researchers should consider all of these factors [159,160]. This is particularly relevant for PRP, a sample with a platelet concentration higher than that normally contained in whole blood [161], and requires a specific preparation method. The standardization of the analytical procedures, including the choice of the matrix and sample pre-treatment, is crucial to reduce the heterogeneity of results.

About the second aim (to verify if the increase in 5-HT levels is associated with specific cognitive, neuro-behavioral, or clinical outcomes), as reported in Table 2, only 26 articles (23%) found a correlation between 5-HT and some clinical outcomes, as particularly evident in patients with higher global severity of ASD. Studies focusing specifically on Asperger’s syndrome or high-functioning ASD generally confirmed the presence of hyperserotonemia in these conditions [58,61,81,97]. However, other studies suggest that elevated 5-HT levels may be prevalent in ASD patients with intellectual disabilities compared to those without (see Table 2 for more details) [24,33,52,77,112]. Notably, there is no evidence of increased 5-HT levels in individuals with idiopathic intellectual disabilities without autism, with some authors reporting reduced 5-HT blood levels [105]. Moreover, some authors found a significant correlation between hyperserotonemia and specific cognitive aspects, such as verbal skills [92] or working memory [112], which are typically impaired in individuals with ASD [162,163]. Also, in this case, it must be considered that diagnostic and clinical assessment instruments varied between studies. For example, some studies relied solely on parental-report measures, while others did not specify the diagnostic criteria and tools used. Also, only 52% of the reviewed studies reported that their investigations excluded secondary causes of autism, such as genetic, neurometabolic, and neuroimaging assessments. Therefore, the risk of inclusion of individuals with secondary forms of autism is present. On the other hand, the selection of control groups was another concern since only a few studies experimentally confirmed that the control participants were free of autistic traits. This is particularly relevant in light of the expanded definition of the autism spectrum [78,164,165] and its growing prevalence [4].

Most studies that reveal a positive correlation between 5-HT and the severity of core symptoms of ASD are recent and have higher quality scores on average, using the CARS as an evaluation tool [166,167]. Therefore, further investigation conducted with greater experimental accuracy could provide a clear association between 5-HT levels and clinical outcomes and determine the potential role of hyperserotonemia as a predictive marker for comorbidities in ASD.

Table 3 summarizes the suggestions emerging from the review in order to minimize heterogeneity and prevent inconsistencies in the results of future studies. Important variables concern the demographic and epidemiological characteristics of patients and control, criteria and tools used to diagnose ASD, and analytical procedures for 5-HT assay.

About ASD subjects, the presence of comorbidities, including medical conditions such as gastrointestinal, neurological, psychiatric, and neurodevelopmental disorders, should also be evaluated. Regarding gastrointestinal disorders, some low-quality articles agree on the presence of a positive correlation between gastrointestinal symptoms and 5-HT levels [16,65]. This could be due to a relevant link between 5-HT, ASD, and microbiome alterations [28,168,169]. However, other studies investigating gastrointestinal symptoms in children with ASD did not find any significant correlation with 5-HT [53,67]. 

Moreover, the impact of diet and medication on 5-HT levels should be considered when interpreting results, as both factors can influence 5-HT levels (and ultimately affect the correlation between hyperserotonemia and ASD). Several dietary sources of exogenous 5-HT or tryptophan (its precursor) could modify its brain and blood concentrations [170,171]. Moreover, it has been shown that fluctuations in serum 5-HT levels are associated with meal intake, exhibiting a reduction during the postprandial period [172]. Similarly, it is well known that some drugs, such as selective serotonin uptake inhibitors (SSRIs), fenfluramine, and other substances, may significantly alter 5-HT blood levels both in autistic and neurotypical subjects [101,137,173]. In addition, the controls’ characteristics are essential to consider; for instance, the relatives of ASD patients are not the best controls, as they may have 5-HT alterations even if they are typically developing.

Another crucial aspect is the frequent use of different diagnostic tools for assessing autism observed in the current work. ADI/ADI-R and ADOS/ADOS-2 are the most frequently used diagnostic tools and have often been described as the “gold standard” for evaluating autism [174]. Nevertheless, it has been observed that the combined ADI-ADOS likelihood of making a correct diagnostic decision about autism is around 68% [175,176]. Regarding the accuracy of the CARS (the third most used tool), a recent meta-analysis [177] showed that, although its internal consistency and sensitivity were considered acceptable, the specificity was not. Nevertheless, it has been shown that CARS and ADOS reports display a high agreement with clinical diagnosis based on DSM criteria [178].

A recent study [179] demonstrated a 787% exponential increase in the recorded incidence of autism diagnoses between 1998 and 2018 in the UK, consistent with other reports from Europe and the United States [180,181]. This increase, due also to greater public awareness of autism, increased assessment demand, as well as augmented attention to populations under-evaluated in the past, including females and adults, requires careful consideration when using diagnostic tools. Although single diagnostic instruments are more cost-effective and less time-consuming, studies pointed to the need for using them at the early stages of the assessment process and then to proceed with a joint administration with other confirmatory tools.

Additionally, the heterogeneity observed in the measurement of 5-HT levels across studies may stem from various factors beyond differences in laboratory procedures and techniques. Variations in participant characteristics, such as age, the presence of comorbidities, or the accurate exclusion of secondary causes of autism, could contribute to inconsistent findings. For instance, the age range of participants in studies may vary widely, potentially influencing 5-HT levels due to developmental changes. Moreover, comorbid conditions, such as gastrointestinal disorders or mood disorders, which are prevalent in individuals with ASD, could confound the relationship between 5-HT levels and ASD symptoms. Furthermore, the inadequate exclusion of secondary causes of autism, such as genetic or metabolic conditions, may result in the inclusion of individuals with heterogeneous etiologies, contributing to the variability in 5-HT levels observed across studies. Additionally, treatment status, including medication usage or dietary interventions, may impact 5-HT levels and introduce further variability in study outcomes. Therefore, it is essential for future research to consider and control for these potential confounding factors to improve the comparability and reliability of findings in the investigation of hyperserotonemia in ASD. 

Similarly, it would be essential to find an agreement on which is the most suitable matrix to use for sampling. For example, it should be considered that plasma is whole blood minus erythrocytes, leukocytes, and thrombocytes (platelets) [182]; however, there are intrinsic difficulties in measuring plasma 5-HT concentrations due to its low levels and the difficulties in preparing the PPP without any platelet contamination [183]. Moreover, the use of international units to express 5-HT concentration should be encouraged.

There is a need to define standardized protocols to ensure consistency of measurements across studies and avoid potential confounders: pre-sampling patient preparation, type of sample, pre-analytical procedures, and analytical procedures should be validated and standardized to ensure the comparability and reproducibility of results across studies.

Investigating the correlation between hyperserotonemia and core symptoms or other clinical outcomes of ASD is difficult. Future studies should separately consider the effects of hyperserotonemia on ASD-core symptoms, neurocognitive and neuropsychological profiles, and emotional–behavioral symptoms. Some studies have suggested that specific symptoms commonly observed in ASD, such as sleep problems, gastrointestinal issues, and autoimmune disorders, may be associated with hyperserotonemia. 

### Limitations and Future Directions

In addressing the limitations of this review on hyperserotonemia in ASD, firstly, it is crucial to recognize the inherent limitations of relying on existing studies, as this review primarily consolidates findings from previously conducted research. As a secondary source of information, it is therefore subject to the biases and constraints present in the primary studies it synthesizes. 

Furthermore, the reliance on peripheral serotonin levels as a biomarker for ASD warrants consideration on its own. While hyperserotonemia is consistently associated with ASD, serotonin levels do not reflect its brain levels and function; moreover, specific transporter gene variants, the gut–microbiome–brain axis, and pharmacological treatments have all been demonstrated to be able to modify this 5-HT blood–brain balance [28,33,34,35,36,37,38,39,40,101,137,173]. Alternative approaches focused on brain serotonin activity (including neuroimaging techniques or the measurement of other serotonin metabolites) were been considered in the present review, even though they could offer valuable insights and could be found to be more accurate biomarkers for ASD diagnosis or treatment monitoring [24,47,55]. The ongoing longitudinal large cohort studies to identify and validate stratification biomarkers for ASD—whose results have not yet been fully available—offer the potential to address some of the limitations observed in individual studies by providing robust data and enhancing the generalizability of findings [184].

Moreover, it should be noted that the present research focused on idiopathic ASD, but further research should also explore in detail the role of serotonin in specific secondary forms of autism spectrum disorders (e.g., fragile X syndrome) [185]. 

Ethical considerations are also crucial in studies exploring the endophenotypes of autism, as they hold significant implications for translational research. Identifying these endophenotypes could potentially assist in the early diagnosis, clinical or therapeutic monitoring, identification of specific therapeutic targets, and delineation of distinct subgroups of individuals with ASD, which may be challenging to differentiate clinically. However, it is important to ensure that participants’ rights to autonomy and well-being are consistently upheld in such research endeavors. Similarly, attention must be paid to the risk of potential stigmatization or discrimination based on the results of diagnostic biomarkers: with the broadening definition of the autism spectrum, it is essential to remember that diagnosis remains primarily clinical, and therefore, research on endophenotypes should not lead to the exclusion of individuals who require care solely based on biomarker results.

## 5. Conclusions

The present systematic review confirms that 5-HT levels are generally higher in patients with ASD, supporting the role of 5-HT blood levels as an endophenotype of ASD. Despite the positive correlation between 5-HT levels and ASD, no clear relationship has emerged about other clinical outcomes. This work highlighted the heterogeneity of diagnostic tools and experimental methods used in studies, which may have impacted the correlation between 5-HT levels and ASD. 

Concerning the measurement methods, the most commonly used are whole blood and PRP, and the results obtained with the two matrices are coherent despite the different parameters analyzed. Nevertheless, a common pre-sampling protocol must be established to ensure accurate results. Thus, further studies are necessary to evaluate 5-HT levels as predictive of comorbidities, which could enable better patient stratification and early intervention. In addition, international standards for defining hyperserotonemia are required to improve the comparability and generalizability of study findings, and such measures should also address defined protocols for biomaterial selection and pre-sampling preparation to ensure the uniformity of measurements across studies and avoid potential confounders. 

To further understand the role of hyperserotonemia in ASD, future research should focus on investigating potential mechanisms that lead to this specific endophenotype. Additionally, it may be essential to explore the potential role of hyperserotonemia in assisting with diagnosis, determining prognosis, and monitoring clinical conditions commonly associated with the autism spectrum.

## Figures and Tables

**Figure 1 brainsci-14-00481-f001:**
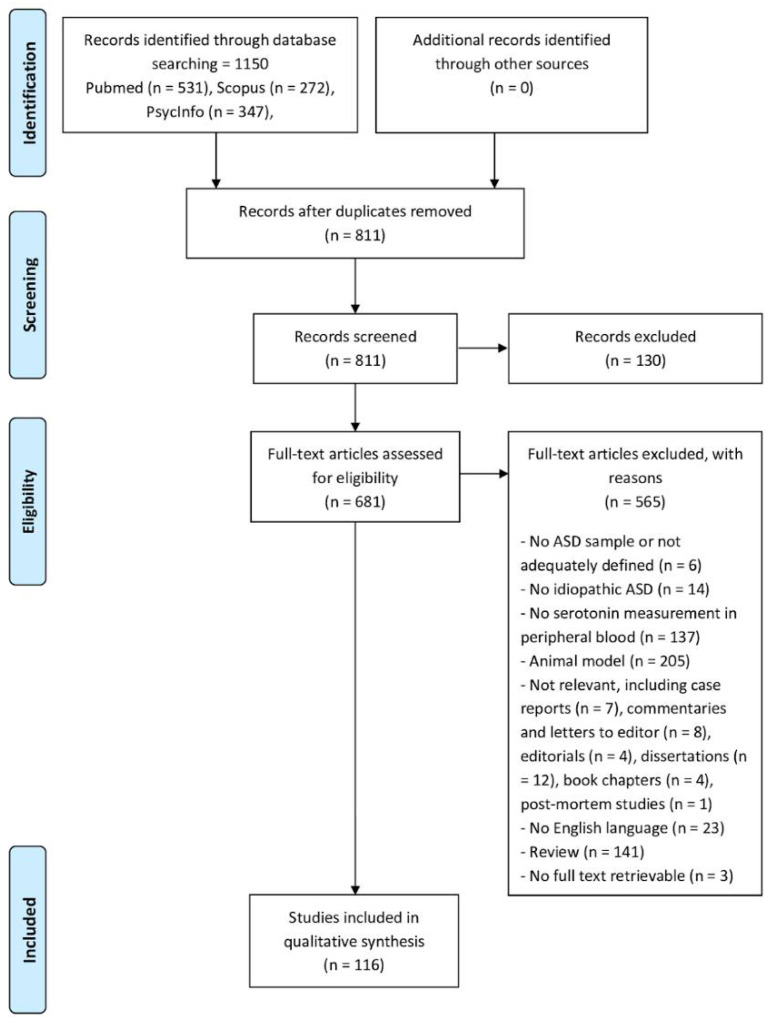
Flowchart of the systematic review, according to PRISMA guidelines.

**Figure 2 brainsci-14-00481-f002:**
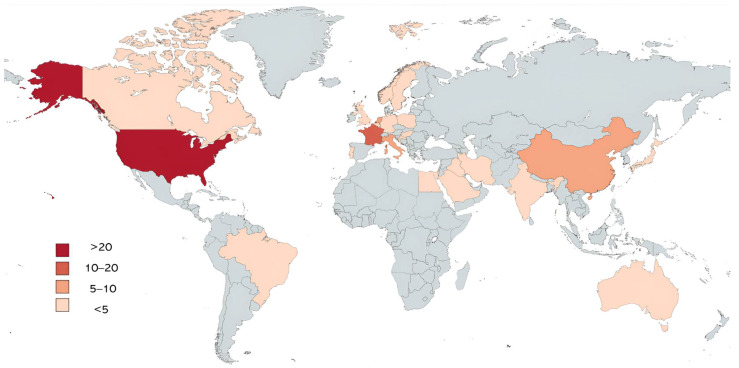
Geographical distribution of the ASD patients studied in the 116 reviewed articles. Countries with a higher number of included articles are represented by darker colors.

**Figure 3 brainsci-14-00481-f003:**
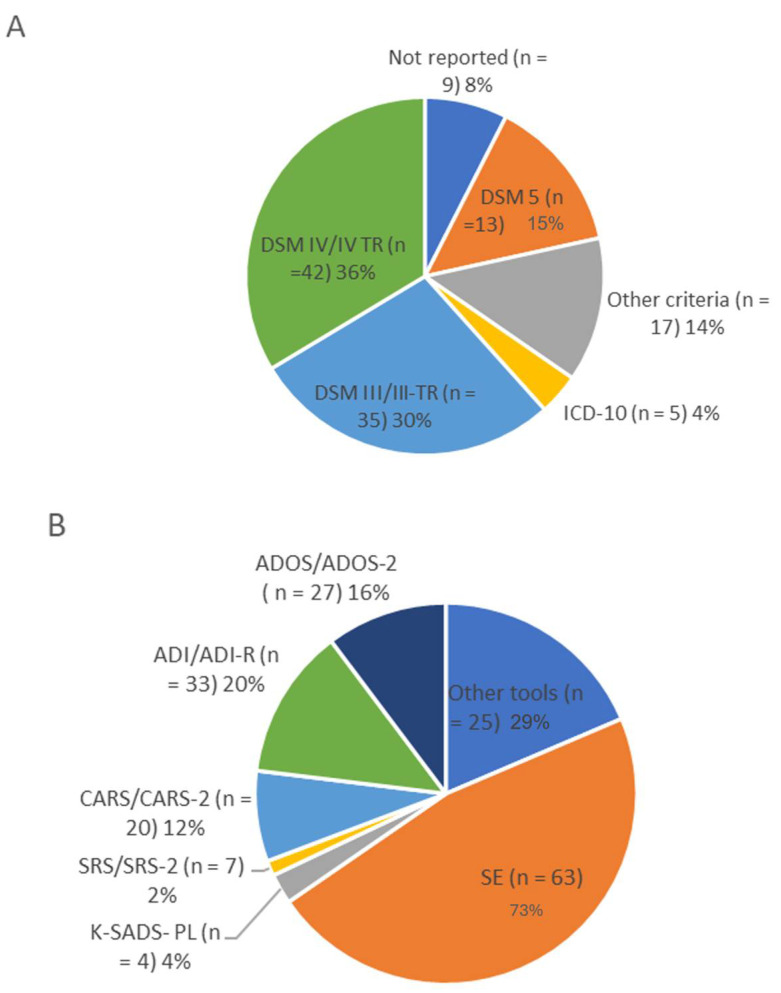
Quantitative synthesis of diagnostic criteria (**A**) and instruments (**B**) used for ASD patients of the studies included in the review. Note that the same study may have applied more than one set of diagnostic criteria or more than one diagnostic instrument, so the total number of papers may be exceeded. ADOS (−2), Autism Diagnostic Observation Schedule (−2); ADI (−R), Autism Diagnostic Interview (—Revised); CARS (−2), Childhood Autism Rating Scale (−2); DSM, Diagnostic and Statistical Manual of Mental Disorders; ICD, International Classification of Diseases; K-SADS-PL, Schedule for Affective Disorders and Schizophrenia for School-Age Children—Present and Lifetime version; SE, Specialist Evaluation; SRS, Social Responsiveness Scale.

**Figure 4 brainsci-14-00481-f004:**
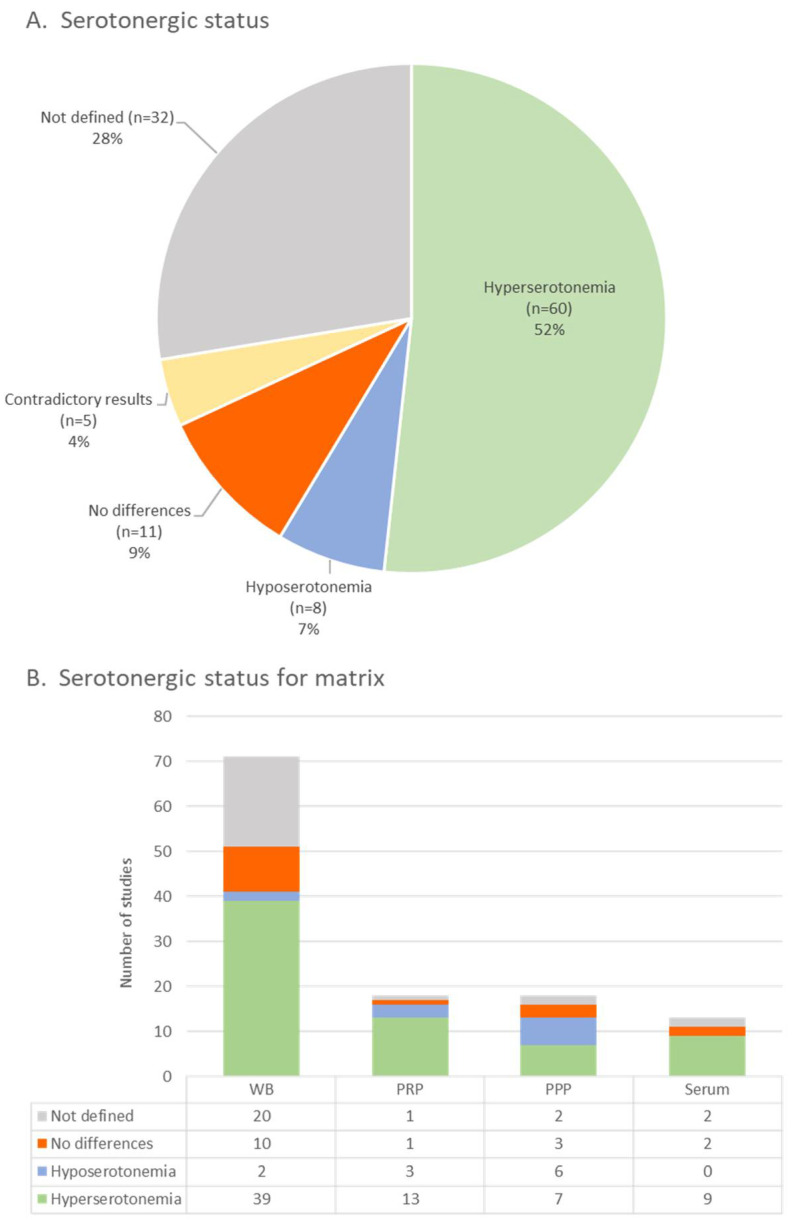
(**A**) Graphic representation of peripheral serotonergic status in ASD patients in 116 reviewed articles. (**B**) Representation of peripheral serotonergic status depending on the used matrix: whole blood (WB), platelet-rich plasma (PRP), platelet-poor plasma (PPP), and serum.

**Table 1 brainsci-14-00481-t001:** Demographic characteristics, country of origin, and diagnostic criteria/tools used in the studies included. Articles are displayed in chronological order, together with quality scores (in parentheses). Both demographic characteristics of autism spectrum disorder (ASD) samples and control groups are reported. Diagnostic criteria and tools used to diagnose ASD and tests administered to healthy controls to confirm typical development (if reported) are also included in this table. ABC = Autism Behavior Checklist; AdAS = Adult Autism Subthreshold Spectrum; ADOS(−2) = Autism Diagnostic Observation Schedule (−2); ADI(−R) = Autism Diagnostic Interview(−Revised); ATEC = Autism Treatment Evaluation Checklist; BAPQ = Broader Autistic Phenotype Questionnaire; CARS = Childhood Autism Rating Scale; Confirmation Tools = tests administered to healthy controls to confirm typical development (when available); Consensus = diagnosis made through consensus of at least two specialists; DSM = Diagnostic and Statistical Manual of Mental Disorders; E-2 = Rimland E-2 Autism Checklist; HC = unrelated healthy controls; ICD = International Classification of Diseases; K-SADS-PL = Schedule for Affective Disorders and Schizophrenia for School-Age Children—Present and Lifetime version; NSAC = National Society for Autistic Children; ODI = Ornitz Developmental Inventory; PAs = parents; PDD-MRS = Scale of Pervasive Developmental Disorder in Mentally Retarded Persons; RAADS = Ritvo Autism and Asperger Diagnostic Scale; RLRS = Ritvo–Freeman Real Life Rating Scale; SE = Specialist evaluation; SDQ = Strengths and Difficulties Questionnaire; SRS = Social Responsiveness Scale; USs = unaffected siblings. Note: Age is reported in years unless otherwise specified. In the case of missing means, median and/or age range are used. Some diagnostic criteria are indicated as bibliographic references.

Reference (Quality Score) [Ref.]	Country	ASD Samples N (Mean Age ± SD), N Females (F)	Criteria/Tools for ASD Diagnosis	Controls N (Mean Age ± SD), N Females (F), [Confirmation Tools]
Carpita et al., 2023 (6) [44]	Italy	24 (27.75 ± 6.97), 7 F	DSM-5/AdAS, RAADS	24 Unaffected relatives (55.42 ± 10.25) 21 F; 24 HC (33.29 ± 8.05), 15 F [AdAS, RAADS]
Cremone et al., 2023 (6) [45]	Italy	24 (27.75 ± 6.97), 7 F	DSM-5/SE	24 Unaffected relatives (55.42 ± 10.25) 21 F; 24 HC (33.29 ± 8.05), 15 F
Xiaoxue, 2023 (5) [46]	China	240 divided into 2 groups: early psychological intervention group N = 120 (4.14 ± 1.15), 50 F; late psychological intervention group N = 120 (4.24 ± 1.24), 51 F	DSM-5/SE	120 HC (4.13 ± 1.12), 60 F
Zuniga-Kennedy et al., 2022 (7) [47]	US (OH)	12 (median: 8, range 4–13), 0 F	DSM-5/ADOS-2	14 HC with gastrointestinal disorder group (median 7.5; range 3–18), 0 F 5 HC without gastrointestinal disorder group (median 13; range 8–18), 0 F [SRS]
Pagan et al., 2021 (5) [24]	France	97 (age and sex not defined)	DSM-IV TR/K-SADS, SRS, ADI-R, ADOS-2	138 PAs (age and sex not defined); 56 USs (age and sex not defined); 106 HC (age and sex not defined) [SRS]
Mostafa et al., 2021 (6) [48]	Egypt	22 (6.11 ± 2.23), 6 F	DSM-5/CARS	22 HC (6.21 ± 2.41), 5 F
Meyyazhagan et al., 2020 (5) [49]	India	98 divided into 2 age groups: Group 1 N = 62 (7.81 ± 2.36), 21 F; Group 2 N = 36 (16.82 ± 3.19) 6 F	DSM-IV TR/SE	98 divided into 2 age groups: Group 1 N = 62 HC (7.76 ± 2.50), 21 F Group 2 N = 36 HC (16.67 ± 1.99), 6 F
Ali et al., 2020 (1) [50]	Iraq	60 (range: 3–8 years), 18 F	NR/SE	28 HC (adults, age not defined), 11 F
Javadfar et al., 2020 (2) [51]	Iran	43 divided into 2 groups: Vitamin D group N = 22 (8.88 ± 2.45), 2 F; Placebo group N = 21 (8.95 ± 3.31) 5 F	DSM-5/CARS, ABC-C, ATEC	-
Chakraborti et al., 2020 (7) [52]	India	104 (5.59 ± 0.37), 17 F	DSM-5/CARS	26 HC (5.13 ± 0.82), 11 F
Wang et al., 2020 (7) [53]	China	26 (4.3; SD not defined), 2 F	DSM-5/SE	24 HC (4.5; SD not defined), 2 F
Hua et al., 2020 (6) [54]	China	120 divided into 2 groups: Sleep disorder group N = 60 (3.99 ± 0.17), 12 F; No sleep disorder group N = 60 (3.93 ± 0.14), 8 F	DSM-5/CARS, ABC, SRS	60 HC (age and sex not defined)
Bridgemohan et al., 2019 (3) [16]	US (MA)	83 (7.4 ± 1.6), 15 F	NR/SE	-
Wichers et al., 2019 (7) [55]	UK	19 (30 ± 11), 0 F	ICD-10/ADI-R, ADOS-2	19 HC (27 ± 9), 0 F
Aaron et al., 2019 (7) [56]	US (IL, TX)	110 (140.7 months ± 73.9), 21 F	DSM-IV TR/ADI-R, ADOS-2	18 HC (201.3 months ± 100.7), 4 F
Montgomery et al., 2018 (5) [57]	US (IL, TX)	188 (9.7 ± 5.2), 31 F	DSM-IV TR/ADI-R, ADOS-2	119 mothers (41.4 ± 6.8), 99 fathers (43.3 ± 7.5)
Lefevre et al., 2018 (6) [58]	France	18 (34.3 ± 7.6), 0 F	DSM-IV TR/ADI	24 HC (26.3 ± 6.3), 0 F
Abdulamir et al., 2018 (6) [59]	Iraq	60 (7.28 ± 2.89), 0 F	DSM-V; ICD-10/SE	26 HC (6.92 ± 2.59), 0 F
Guo et al., 2018 (8) [60]	China	33 (5.14 ± 1.33), 5 F	DSM-5/CARS, ABC	32 HC (5.18 ± 0.87), 6 F
Ormstad et al., 2018 (4) [61]	Norway	65 (11.2, SD not defined), 13 F	ICD-10/ADI-R, ADOS-2	30 HC (10.9, SD not defined), 16 F
Shuffrey et al., 2017 (3) [62]	US (IL)	264 divided into 2 groups: Pre-pubertal group N = 182 (7.56 ± 2.41), 23 F; Post-pubertal group N = 82 (16.72 ± 4.72), 14 F.	DSM-IV TR/ADI-R, ADOS-2	-
Pagan et al., 2017 (5) [63]	France	239 (age and sex not defined)	DSM-IV TR/K-SADS, SRS, ADI-R, ADOS-2	303 PAs (age and sex not defined), 78 USs (age and sex not defined), 278 HC (age and sex not defined) [SRS]
Benabou et al., 2017 (5) [64]	France	213 (14.4 ± 9.1), 39 F	DSM-IV TR/K-SADS, SRS, ADI-R, ADOS-2	364 PAs (46.5 ± 9.5), 128 USs (14.8 ± 8.0), 185 F [SRS]
Chen et al., 2017 (4) [26]	US (TN)	116 (9.67 ± 5.93), 18 F	DSM-IV TR/ADI-R, ADOS-2	63 mothers (age not defined); 72 fathers (age not defined) [BAPQ]
Marler et al., 2016 (3) [65]	US (MO)	82 divided into 2 groups: normoserotonemia group N = 63 (11.2 ± 4.1), 6 F; hyperserotonemia group N = 19 (11.7 ± 3.6), 2 F.	DSM-IV/ADOS-2	-
Chakraborti et al., 2016 (5) [66]	India	203 (range 1.9–14), 33 F	DSM-IV TR or DSM-5/CARS	236 HC (range 4–31), 111 F
Kheirouri et al., 2016 (7) [67]	Iran	35 (8.1 ± 4.0), 11 F	DSM-IV TR/GARS	31 HC (7.3 ± 2.6), 13 F [SDQ]
Francis et al., 2016 (3) [68]	US (IL)	207 (9.88 ± 5.48), 37 F	DSM-IV TR/ADI-R, ADOS-2	-
Chugani et al., 2016 (4) [69]	US (MI, OH, TX, NY)	166 divided into 3 groups: 2.5 mg buspirone group N = 54 (age range 2–6), 8 F; 5.0 mg buspirone group N = 55 (age range 2–6), 11 F, Placebo group N = 57 (age range 2–6), 10 F	DSM-IV TR/ADI-R, ADOS-2	-
Gebril et al., 2015 (7) [70]	Egypt	20 (7.4 ± 2.6), 0 F	DSM-IV/CARS	20 HC (9 ± 1.6). 0 F
Bijl et al., 2015 (7) [71]	Belgium	159 (11.9 ± 3.8), 34 F	DSM-IV TR/SRS	186 PAs (43.0 ± 4.4), 99 F; 103 USs (13.0 ± 4.7), 69 F; 65 pediatric HC (15.6 ± 3.9), 34 F; 45 adult HC (36.5 ± 10.9), 23 F
Jaiswal et al., 2015 (7) [33]	India	169 (5.86 ± 0.24), 28 F	DSM-IV TR/CARS	317 PAs (25.10 ± 1.04), 164 F; 168 HC (21.92 ± 1.15), 87 F
Yang et al., 2015 (6) [72]	China	33 (12.21 ± 2.67), 6 F	DSM-IV/CARS	31 HC (12.52 ± 2.14), 7 F
Yang et al., 2015b (8) [73]	China	43 (7.51 ± 1.47), 8 F	DSM-5/CARS	40 HC (7.83 ± 1.63), 10 F
Alabdali et al., 2014 (7) [74]	Saudi Arabia	52 (7.0 ± 2.34), 0 F	DSM-IV/CARS, SRS	30 HC (7.2 ± 2.14), 0 F
Kolevzon et al., 2014 (3) [75]	US (NY)	64 (6.85 ± 2.75), 19 F	DSM-IV/ADI-R	-
Gabriele, Lombardi, et al., 2014 (6) [76]	Italy	428 (8.86 ± 0.29), 57 F	DSM-IV/ADI-R, ADOS-2	809 PAs (age and sex not defined); 158 USs (age and sex not defined)
Pagan et al., 2014 (8) [77]	France	278 (N = 135 < 16 years; N = 143 > 16 years), 53 F	DSM-IV TR/K-SADS, SRS, ADI-R, ADOS-2	377 PAs (N = 377 > 16 y), 199 F; 129 USs (N = 72 < 16 y, N = 57 > 16 y), 67 F; 416 HC (N = 111 < 16 y, N = 305 > 16 y), 182 F [SRS]
Levin-Decanini et al., 2013 (5) [78]	US (IL)	197 (240 ± 67.32 months), 35 F	DSM-IV TR/ADI-R, ADOS-2	196 mothers (492 ± 84 months); 161 fathers (532 ± 96 months) [BAPQ]
Anderson et al., 2012 (7) [79]	US (NY)	18 (10.1 ± 5.5), 0 F	DSM-III-R/SE	24 HC (14.2 ± 8.5), 4 F
Mostafa & Al-Ayadhi, 2011 (8) [80]	Saudi Arabia	50 (8.22 ± 2.28), 9 F	DSM-IV/CARS	30 HC (8.23 ± 2.36), 5 F
El-Ansary et al., 2011 (6) [81]	Saudi Arabia	25 (range 4–12), sex not defined	NR/ADI-R, ADOS, 3DI	16 HC (range 4–11), sex not defined
Sacco et al., 2010 (3) [82]	Italy	245 (8.82 ± 5.62), 29 F	DSM-IV/ADI-R, ADOS	-
Kazek et al., 2010 (8) [83]	Poland	51 (8,1, SD not defined), 12 F	NR/SE	28 HC (7.9, SD not defined), 8 F
Kolevzon et al., 2010 (3) [84]	US (NY)	78 (6.77 ± 2.93), 13 F	ICD-10, DSM-IV/ADI-R	-
Mulder et al., 2009 (7) [85]	The Netherlands	19 divided into 2 groups: normoserotonemic N = 10 (15.3 ± 4.0), 0 F; Hyperserotonemic N = 9 (15.3 ± 4.4), 0 F	DSM-IV TR/ADI-R, ADOS	-
Kemperman et al., 2008 (3) [86]	The Netherlands	24 (9.9 ± 3.9), 6 F	DSM-IV TR/SE	-
El-Sherif et al., 2008 (8) [87]	Egypt	40 (7.35 ± 2.6), 8 F	DSM-IV/CARS	40 HC (7.68 ± 2.5), 8 F
Warren & Singh, 2008 (7) [88]	US (UT)	20 (10.1, SD not defined), 4 F	DSM-III-R/SE	13 HC (8.8, SD not defined), 3 F
Sacco, Papaleo, et al., 2007 (6) [89]	Italy	371 (age and sex not defined)	DSM-IV/ADI-R, ADOS	156 USs (age and sex not defined); 180 HC (age and sex not defined)
Sacco, Militerni, et al., 2007 (3) [90]	Italy	241 (7.10 ± 2.78), 36 F	DSM-IV/ADI-R, ADOS	-
Coutinho et al., 2007 (7) [91]	Portugal	186 (6.8, SD not defined), sex not defined	DSM-IV/ADI-R, CARS	181 adult HC, (age and sex not defined)
Hranilovic et al., 2007 (7) [92]	Croatia	53 (26.1 ± 6.6), 15 F	DSM-IV/CARS	45 HC (39.2 ± 9.2), 1 F
Connors et al., 2006 (7) [93]	US (TN)	17 (age range 2–18), 0 F	DSM-IV/CARS	17 mothers (range 25–40); 12 fathers (range 27–45); 7 USs (range 6–15, sex not specified); 8 mothers of HC (age range 25–40)
Weiss et al., 2006 (6) [94]	US (IL, TN)	50 (5.9 ± 3.3), sex not defined	DSM-IV/ADI-R, ADOS	567 Hutterites (age and sex not defined); 392 HC adults (age and sex not defined)
Croonenberghs et al., 2005 (7) [95]	Belgium	18 (16.2 ± 1.7), 0 F	DSM-IV/ADI-R, Consensus	22 HC (16.0 ± 1.8), 0 F
Spivak et al., 2004 (6) [96]	Israel	10 (24.3 ± 4.5), 4 F	DSM-IV/CARS, RLRS	12 HC (30.0 ± 3.6), 6 F
Mulder et al., 2004 (8) [97]	The Netherlands	81 divided into 3 groups: autism N = 33 (11.7 ± 4.0), 4 F; Asperger’s N = 5 (13.2 ± 3.3), 1 F; PDD-NOS N = 43 (13.1 ± 4.2), 6 F	DSM-IV-TR/ADI-R, ADOS, ABC, PDD-MRS	54 Cognitively impaired (13.0 ± 3.3), 11 F; 60 HC (11.5 ± 3.9), 31 F [ABC, PDD-MRS]
Coutinho et al., 2004 (5) [34]	Portugal	105 (7.14, SD not defined), sex not defined	DSM-IV/ADI-R, CARS	52 HC (7.27, SD not defined), sex not defined
Martin et al., 2003 (3) [98]	US (CT)	18 (11.3 ± 3.6), 4 F	NR/ADI-R, ADOS	-
Vered et al., 2003 (7) [99]	Israel	7 (25.4 ± 4.8), 3 F	DSM-IV/Consensus	10 HC (30.3 ± 4.6), 5 F
Betancur et al., 2002 (5) [100]	France	150 (14.5, SD not defined), 41 F	DSM-IV/ADI-R	PA and USs (N, age and sex not defined)
Persico et al., 2002 (6) [101]	Italy	134 (age and sex not defined)	DSM-IV/ADI-R, ADOS	244 PAs (age and sex not defined), 49 USs (age and sex not defined)
Croonenberghs et al., 2000 (7) [102]	Belgium	13 (14.5 ± 1.8), 0 F	DSM-IV/Consensus	13 HC (15.1 ± 1.5), 0 F
Leboyer et al., 1999 (8) [103]	France	62 (9.2 ± 4.2), 20 F	DSM-III-R, ICD-10/ADI	61 mothers (age not defined), 42 fathers (age not defined), 11 sisters (age not defined); 9 brothers (age not defined), 91 young HC (range 2–16; sex not defined), 118 adult HC (age and sex not defined)
McBride et al., 1998 (8) [104]	US (NY)	7 (23.1 ± 3.8), 0 F	DSM-III-R/ODI, ABC, ADOS	8 HC (25 ± 2.8), 0 F
Singh et al., 1997 (8) [105]	US (UT)	23 (6.3, SD not defined), 5 F	DSM-III-R/Consensus	10 Cognitively impaired (6.5, SD not reported), 4 F; 23 HC (6.8, SD not reported), 11 F
Hérault et al., 1996 (3) [106]	France	65 (7.0, SD not defined), 25 F	DSM-III-R/Consensus	-
Bouvard et al., 1995 (3) [107]	France	10 (9.5, SD not defined), 5 F	DSM-III-R/ADI	-
Tordjman et al., 1995 (9) [108]	US (NY)	38 (prepubertal 5.5 ± 2.49; postpubertal 22.9 ± 4.8), 0 F	DSM-III-R/ODI, ABC, ADOS	12 Cognitively impaired (6.00 ± 2.45), 0 F; 21 HC (prepubertal 7.70 ± 2.45, postpubertal 22.1 ± 5.8), 0 F
László et al., 1994 (5) [109]	Hungary	46 (5.4, SD not defined), 9 F	DSM-III/SE	20 HC (range 2–10), sex not defined
Rolf et al., 1993 (7) [110]	Germany	18 (9.9 ± 2.8), 2 F	DSM-III/SE	14 HC (11.5 ± 2.0), 6 F
Naffah-Mazzacoratti et al., 1993 (6) [111]	Brasil	19 (range 1–12), sex not defined	DSM-III-R/SE	46 HC (range 1–12), sex not defined
Cuccaro et al., 1993 (6) [112]	US (SC)	18 (18 ± 9), 5 F	DSM-III-R/SE	21 PAs (45 ± 12), 13 F; 13 USs (24 ± 13),4 F
Leventhal et al., 1993 (3) [113]	US (IL)	15 (7.6 ± 2.6), 2 F	DSM-III/ODI, Consensus	-
Hérault et al., 1993 (7) [114]	France	23 (5.9, SD not defined), 7 F	DSM-III-R/SE	59 HC (6.0, SD not defined), 25 F
Yuwiler et al., 1992 (6) [115]	US (CA)	N not defined (10, SD not defined), sex not defined	DSM-III-R, NSAC/SE	Obsessive-compulsive patients (N not defined, mean age 13, SD not defined), sex not defined; multiple sclerosis patients (N not defined, mean age 41, SD not defined), sex not defined; HC (N not defined, mean age 33, SD not defined), sex not defined
Duker et al., 1991 (2) [116]	The Netherlands	11 (18.36 ± 8.38), 4 F	DSM-III-R/consensus	-
Perry et al., 1991 (8) [117]	US (IL)	12 (7.5 ± 2.9), 0 F	DSM-III-R/consensus	22 PAs (35.2 ± 4.1), 11 F; 6 USs (8.3 ± 1.8), 2 F; 7 adult HC (24, SD not reported), 0 F; 10 child HC (11.0 ± 2.7), 0 F.
Piven et al., 1991 (7) [118]	US (MD)	28 divided into 2 groups: Simplex N = 23 (21.5, SD not defined), 4 F; Multiplex N = 5 (26.4, SD not defined), 0 F	DSM-III-R/ADI	10 HC (25.1, SD not reported), 5 F
Stern et al., 1990 (3) [119]	Australia	20 (mean age 10.0, SD not defined), 6 F	DSM-III/SE	-
Cook et al., 1990 (7) [120]	US (IL)	16 (9.0 ± 3.5), 0 F	DSM-III-R/consensus	53 PAs (37.3 ± 5.1), 29 F; 21 USs (11.9 ± 5.4), 9 F
Oades et al., 1990 (2) [121]	Australia	7 (11.3 ± 4.0), 1 F	DSM-III/SE	-
Leventhal et al., 1990 (6) [122]	US (IL)	39 (8.99 ± 4.37), 7 F	DSM-III-R/SE	78 PAs (38.71 ± 6.52), 42 F; 32 USs (11.22 ± 5.27), 18 F
Abramson et al., 1989 (8) [123]	US (SC)	57 (13.9 ± 5), 4 F	DSM-III/SE	17 HC (13.3 ± 6.5), 14 F
Ekman et al., 1989 (3) [124]	Sweden	20 (6.25, SD not defined), 2 F	DSM-III-R; Rutter (1978)/ABC, RLRS	-
McBride et al., 1989 (8) [125]	US (NY)	7 (23.1 ± 3.8), 0 F	DSM-III-R/ABC, ODI, consensus	8 HC (25.0 ± 2.8), 0 F
Sherman et al., 1989 (2) [126]	Canada	15 (11.4, SD not defined), 2 F	DSM-III; NSAC (1981)/RLRS	-
Minderaa et al., 1989 (8) [127]	The Netherlands	40 (19.4 ± 4.9), 12 F	DSM-III/SE	20 HC (22.0 ± 7.5), 5 F
Coggins et al., 1988 (1) [128]	US (CA)	5 (age not defined), 1 F	NR/consensus	-
Cook et al., 1988 (4) [129]	US (IL)	22 (10.7 ± 3.4), 1 F	NR/SE	21 mothers (37.7 ± 4.5), 14 fathers (41.2 ± 5.2), 9 brothers (10.5 ± 5.1), 8 sisters (15.0 ± 6.3)
Geller et al., 1988 (8) [130]	US (CA)	19 (range 20–140 months), sex not defined	DSM-III/Consensus	6 with schizophrenic reaction in childhood (range 20–140 months), sex not defined; 26 HC (range 20–140 months), sex not defined
Launay et al., 1988 (7) [131]	France	22 (10.5, SD not defined), 6 F	DSM-III/SE	22 HC (age and sex not defined)
Kuperman et al., 1987 (3) [132]	US (IA)	25 (126.2 ± 52.6 months), 0 F	DSM-III/ABC	-
Badcock et al., 1987 (7) [133]	Australia	30 (10.2, SD not defined), sex not defined	DSM-III; NSAC (1981)/SE	11 developmental dysphasia (5.5, SD not defined), sex not defined; 106 HC (9.7, SD not defined), sex not defined
Launay et al., 1987 (7) [134]	France	22 (10.5, SD not defined), sex not defined	DSM-III/SE	HC (N, age and sex not defined)
Minderaa et al., 1987 (8) [135]	The Netherlands	36 divided into 2 groups: unmedicated N = 16 (20.6 ± 4.6), 5 F; medicated N = 20 (19.4 ± 4.1), 4 F	DSM-III/SE	27 HC (20.3 ± 6.9), 8 F [self-report questionnaire]
Anderson et al., 1987 (7) [136]	US (CT)	40 (16.8 ± 6.04), 11 F	DSM-III/SE	87 HC (14.6 ± 7.47), 42 F [pediatrician evaluation and parental questionnaires]
Ho et al., 1986 (6) [137]	Canada	31 (8, SD not defined), sex not defined	DSM-III/SE	10 with cognitive impairment (7, SD not defined), sex not defined; 18 Down syndrome (9, SD not defined), sex not defined; 23 HC (10, SD not defined), sex not defined
Stubbs et al., 1986 (1) [138]	US (OR)	8 (7.88 ± 4.26), 4 F	DSM-III/SE	-
Israngkun et al., 1986 (8) [139]	US (OH)	14 (14.57 ± 5.64), 3 F	DSM-III, Rimland (1968)/CARS, E-2	10 HC (14.6 ± 5.08), 2 F
Piggott et al., 1986 (1) [140]	US (MI)	8 (range 4–15), sex not defined	DSM-III; NSAC/SE	-
August et al., 1985 (1) [141]	US (TX)	9 (8.44 ± 2.40), 1 F	NR	-
Kuperman et al., 1985 (6) [142]	US (IA)	25 males (10.5 ± 4.3), and 5 F (9.9 ± 5.0)	DSM-III/SE	30 mothers (36.2 ± 7.2); 24 fathers (38.3 ± 8.3); 11 brothers (9.0 ± 4.9); 10 sisters (13.2 ± 9.8)
Hoshino et al., 1984 (6) [143]	Japan	37 (4.7, SD not defined), 3 F	Kanner (1943), Rutter (1972)/E-2	12 young HC (10.5, SD not defined), 6 F; 28 adult HC (26.8, SD not defined), 17 F
August et al., 1984 (7) [144]	US (TX)	10 (range 5–12), sex not defined	DSM III; NSAC/SE	8 HC (age and sex not defined)
Ritvo et al., 1984 (3) [145]	US (CA)	14 (range 2–18), 3 F	DSM-III; NSAC/RLRS	-
Ritvo et al., 1983 (3) [146]	US (CA)	14 (range 2–12), 3 F	DSM-III; NSAC/RLRS	-
Rotman et al., 1980 (1) [147]	Israel	4 (range 8–14), sex not defined	NR/SE	-
Hanley et al., 1977 (6) [21]	US (IL)	27 (age and sex not defined)	Schain and Freedman criteria/SE	25 severe cognitive impairment (age and sex not defined); 23 mild cognitive impairment (age and sex not defined); 6 HC (age and sex not defined)
Takahashi et al., 1977 (7) [148]	Japan	20 (4.5 ± 2.9), 1 F	Kanner (1943)/consensus	39 psychiatric and neurological patients (7.5 ± 3.4), 17 F; 30 HC (5.4 ± 3.2), 7 F
Takahashi et al., 1976 (8) [149]	Japan	30 (4.8 ± 2.9), 3 F	Kanner (1943)/consensus	45 psychiatric and neurological patients (7.1 ± 3.3), 20 F; 30 HC (5.4 ± 3.2), 7 F
Yuwiler et al., 1975 (6) [150]	US (CA)	12 (60.2 ± 31.1 months), 1 F	Kanner (1943)/consensus	15 hospitalized patients (69.4 ± 20.6 months), 7 F; HC (N not defined, 97.5 ± 25.9 months), sex not defined
Ritvo, 1971 (4) [151]	US (CA)	4 (range 3–13), 0 F	Rimland (1968)/consensus	4 non-autistic patients (age and sex not defined)
Yuwiler et al., 1971 (3) [152]	US (CA)	7 (60.6 ± 24.2 months), 0 F	Rimland (1968)/consensus	4 non-autistic patients (71.5 ± 36.9 months), 1 F
Ritvo et al., 1970 (6) [153]	US (CA)	24 (58.2 ± 15.8 months), 6 F	Kanner (1943)/consensus	HC (N, age and sex not defined)

**Table 2 brainsci-14-00481-t002:** List of studies reporting correlation between clinical outcomes and hyperserotonemia in ASD patients. Twenty-six studies (23%) reporting correlation (positive or negative) are reported. Studies that did not report any specific correlation between clinical outcomes and serotonemia are omitted. ↑ = positive correlation with hyperserotonemia; ↓ = negative correlation with hyperserotonemia; Correlation = correlation with serotonin blood levels; QA = quality assessment of the referenced study.

Correlation	References (QA)
**Autism severity**	
↑	Leventhal et al., 1993 [113] (3); Mostafa et al., 2011 [80] (8); Alabdali et al. 2014 [74] (7); Yang et al., 2015 [72] (6); Yang et al., 2015 [73] (8); Marler et al., 2016 [65] (3); Abdulamir et al., 2018 [59] (6); Guo et al.,2018 [60] (8); Chakraborti et al., 2020 [52] (7); Pagan et al., 2021 [24] (5); Mostafa et al., 2021 [48] (8)
↓	Cook et al., 1990 [120] (7); Sacco et al., 2010 [82] (3); Jaiswal et al. 2015 [33] (7); Kheirouri et al., 2016 [67] (7); Chakraborti et al., 2016 [66] (5); Ormstad 2018 [61] (4); Pagan et al., 2021 [24] (5)
**Intellectual disability**	
↑	Pagan et al., 2021 [24] (5)
**Repetitive behaviors**	
↑	Sacco et al., 2010 [82] (3); Yang et al., 2015 [72] (8); Marler et al., 2016 [65] (3).
↓	Pagan et al., 2021 [24] (5)
**Verbal communication**	
↓	Hranilovic et al., 2007 [92] (7)
**Memory**	
↓	Cuccaro et al., 1993 [112] (6)
**Self-injurious behaviors**	
↑	Yang et al., 2015 [73] (8)
↓	Sacco et al., 2010 [82] (3); Kolevzon et al., 2010 [84] (3) and 2014 [75] (3)
**Hetero-aggressive behaviors**	
↑	Spivak et al., 2004 [96] (6)
**Gender differences in emotion expression**	
↑	Chakraborti et al., 2020 [52] (7)
**Sleep problems**	
↑	Hua et al., 2020 [54] (6)
↓	Kheirouri et al., 2016 [67] (7)
**Gastrointestinal problems**	
↑	Marler et al., 2016 [65] (3); Bridgemohan et al., 2019 [16] (3)
**Allergies and immune disorders**	
↑	Sacco et al., 2010 [82] (3); Mostafa et al., 2021 [48] (8)

**Table 3 brainsci-14-00481-t003:** Essential variables to consider in planning studies on 5-HT and ASD.

**Patients:** Report demographic and epidemiological characteristics (ethnicity, gender, age, and pubertal stage). Pay attention to medications, drugs, and diets that can affect 5-HT metabolism. Evaluate and report comorbidities: − Internal medical conditions and known medical conditions (e.g., gastrointestinal disorders); − Other neurological (including seizures and sleep disturbances), psychiatric (e.g., mood disorders and psychosis), and neurodevelopmental (e.g., intellectual disability and ADHD) disorders; − Screening for genetic and metabolic conditions associated with ASD. Criteria and diagnostic tools for the diagnosis of ASD: avoid single diagnostic instruments and favor administration of confirmatory tools. Clinical outcome: − Separately consider the effects on ASD-core symptoms, neurocognitive and neuropsychological profiles, and emotional–behavioral symptoms.
**Controls:** Avoid relatives of subjects with ASD as controls (having often altered 5-HT levels). Screen for ASD symptoms (given the high prevalence of ASD).
**Peripheral 5-HT assay:** Pay attention to pre-analytical procedures (e.g., blood sampling, storage, and sample pre-treatment). Biomaterials analyzed (e.g., whole blood, platelet-rich plasma, and platelet-poor plasma): prevent analytical problems and contamination risks (consider the intrinsic difficulties to measure plasma 5-HT concentrations due to its low levels; be aware of the platelet contamination risks in preparing PPP). Report analytical method (e.g., HPLC and ELISA).

## Data Availability

No new data were created or analyzed in this study. Data sharing is not applicable to this article.

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
