# Peer review of "A Systematic Review on Autism and Hyperserotonemia: State-of-the-Art, Limitations, and Future Directions"

_brainsci, 2024, doi:10.3390/brainsci14050481_

Round 1

Reviewer 1 Report

Comments and Suggestions for Authors

This is an important topic because investigating hyperserotonemia in ASD holds promise for improving our understanding of the disorder, leading to better diagnosis, treatment strategies, and potentially even prevention methods. This systematic review appears to be well-conducted overall. Here are some of its strengths:

  • The introduction outlines three clear goals for the review: verifying hyperserotonemia as an ASD endophenotype, identifying clinical outcomes in hyperserotonemic ASD patients, and recommending ways to improve research.
  • The authors followed established standards for conducting and reporting systematic reviews by registering their protocol and adhering to PRISMA guidelines.
  • The methods section details the databases searched, search terms used, and date range for the literature search.
  • Defined criteria for selecting relevant studies enhance the objectivity of the review process.
  • The authors describe extracting various data points from the included studies and using a modified Newcastle-Ottawa Scale to assess their quality.

The Review goals are:

  • Verify the consistency of hyperserotonemia as an ASD endophenotype.
  • Identify clinical outcomes associated with elevated serotonin levels in ASD patients.
  • Recommend ways to improve future research on this topic.

Introduction:

The introduction section effectively sets the stage for the rest of the review and highlights the importance of investigating hyperserotonemia in ASD.

Suggestions for Improvement:

I suggest to provide more details on data extraction:

While the authors mention extracting information on "evaluated outcomes," specifying some examples (e.g., specific measures of serotonin levels, and co-occurring conditions) would provide a clearer picture of the data analyzed.

I also recommend elaborating on the quality assessment tool:

Briefly mentioning the specific modifications made to the Newcastle-Ottawa Scale (if relevant) or referencing the specific version used could enhance transparency.

I also suggest addressing the limitations of the study design:

The review itself is a secondary source of information, so acknowledging the limitations of relying on existing studies would strengthen the overall analysis.

I also think that considering future research directions is important:

In addition to recommendations for removing experimental obstacles, the review could discuss broader directions for future research on hyperserotonemia and ASD.

Overall, the introduction and methodology appear to be a well-structured and informative section in this systematic review. 

The suggested improvements would provide even more clarity and comprehensiveness to the sections.

Results:

The results section is very informative but I think that the level of presented information is possible to be summarized. The level of detail provided in the results section can be a double-edged sword. The benefit is that a comprehensive presentation allows readers to understand the data thoroughly and assess the quality of the evidence. However, the drawback is that an excessive amount of detail can make the results section difficult to read and navigate.

My suggestions for improvements are:

Consider summarizing some of the repetitive information about study characteristics in the text, and referring readers to the tables and figures for details.

For the section on serotonin analysis, I suggest focusing on key findings and trends, especially regarding inconsistencies across studies (e.g., different units of measurement, and variations in pre-study treatments).

Consider placing highly technical details about analytical methods in a supplementary file.

Discussion

The discussion section effectively summarizes the key findings and proposes future research directions. I suggest the following points to be considered in the section:

Possible explanations for heterogeneity: The discussion could delve deeper into possible reasons for the inconsistent findings across studies, such as variations in participant characteristics (age, comorbidities) or treatment status.

Addressing limitations of serotonin as a biomarker: The authors mention that serotonin levels are not a perfect indicator of brain serotonin function. They could discuss alternative approaches to measuring brain serotonin activity.

Ethical considerations: Future research on potential uses of serotonin levels in ASD diagnosis or treatment should consider ethical implications.

Author Response

Dear Editor, dear Reviewers,

thank you for taking the time to revise our manuscript, which now appears to us to be significantly improved. The detailed responses to the reviewers' comments are provided below. Changes are highlighted in bold in the main manuscript.

Reviewer #1

Comment #1: This is an important topic because investigating hyperserotonemia in ASD holds promise for improving our understanding of the disorder, leading to better diagnosis, treatment strategies, and potentially even prevention methods. This systematic review appears to be well-conducted overall. Here are some of its strengths:

The introduction outlines three clear goals for the review: verifying hyperserotonemia as an ASD endophenotype, identifying clinical outcomes in hyperserotonemic ASD patients, and recommending ways to improve research.

The authors followed established standards for conducting and reporting systematic reviews by registering their protocol and adhering to PRISMA guidelines.

The methods section details the databases searched, search terms used, and date range for the literature search.

Defined criteria for selecting relevant studies enhance the objectivity of the review process.

The authors describe extracting various data points from the included studies and using a modified Newcastle-Ottawa Scale to assess their quality.

The Review goals are:

Verify the consistency of hyperserotonemia as an ASD endophenotype.

Identify clinical outcomes associated with elevated serotonin levels in ASD patients.

Recommend ways to improve future research on this topic.

Authors’ Reply: We thank the Reviewer for his/her comments. We are pleased that the Reviewer found our manuscript interesting and has highlighted its strengths, we appreciate it.

Introduction:

Comment# 2: The introduction section effectively sets the stage for the rest of the review and highlights the importance of investigating hyperserotonemia in ASD. Suggestions for Improvement: I suggest to provide more details on data extraction: While the authors mention extracting information on "evaluated outcomes," specifying some examples (e.g., specific measures of serotonin levels, and co-occurring conditions) would provide a clearer picture of the data analyzed.

Authors’ reply: We welcomed the Reviewer’s suggestion, and the following sentence has been added to provide more details on data extraction (page 4): “f) evaluated outcomes (i.e., measures of 5-HT levels, correlations between 5-HT levels and ASD-core symptoms, neurocognitive, emotional-behavioural or other relevant clinical outcomes in subjects with ASD)”.

Comment #3: I also recommend elaborating on the quality assessment tool: Briefly mentioning the specific modifications made to the Newcastle-Ottawa Scale (if relevant) or referencing the specific version used could enhance transparency.

Authors’ reply: We agree with the Reviewer’s opinion that elaborating on the quality assessment process can foster transparency. However, due to length limitations, we could not provide a full description of the quality assessment procedure in the manuscript. For this reason, when we wrote the manuscript, we opted to include a full description of the modified Newcastle-Ottawa Scale (and its items) employed in the present review, as well as a detailed description of the quality assessment procedure (including a full scoring for each of the included articles), in the supplementary material section. We agree that this information could have been clearer in the main manuscript, thus we added the following sentence in the method section (page 4): “A full description of the modified NOS employed and on the quality assessment procedure is available in Table 1S and Table 2S”.

Comment #4: I also suggest addressing the limitations of the study design: The review itself is a secondary source of information, so acknowledging the limitations of relying on existing studies would strengthen the overall analysis.

Authors’ reply: We welcomed the Reviewer’s suggestion and added a “Limitations and Future Directions” sub-section at the end of the discussion where we addressed this and other limitations (see also Comment #5, #10 and #11).

Comment #5: I also think that considering future research directions is important: In addition to recommendations for removing experimental obstacles, the review could discuss broader directions for future research on hyperserotonemia and ASD.

Authors’ reply: See previous reply.

Comment # 6: Overall, the introduction and methodology appear to be a well-structured and informative section in this systematic review. The suggested improvements would provide even more clarity and comprehensiveness to the sections.

Authors’ reply: We thank again the Reviewer for his/her valuable comment.

Results:

Comment #7: The results section is very informative but I think that the level of presented information is possible to be summarized. The level of detail provided in the results section can be a double-edged sword. The benefit is that a comprehensive presentation allows readers to understand the data thoroughly and assess the quality of the evidence. However, the drawback is that an excessive amount of detail can make the results section difficult to read and navigate.

My suggestions for improvements are: Consider summarizing some of the repetitive information about study characteristics in the text, and referring readers to the tables and figures for details.

Authors’ reply: We agree with the Reviewer’s suggestion, therefore we removed most of the 3.3.4 paragraph (Peripheral serotonin concentration) from the main text, including its content as a Supplementary Table 3S, in order to improve readability.

Comment #8: For the section on serotonin analysis, I suggest focusing on key findings and trends, especially regarding inconsistencies across studies (e.g., different units of measurement, and variations in pre-study treatments). Consider placing highly technical details about analytical methods in a supplementary file.

Authors’ reply: See previous reply.

Discussion. The discussion section effectively summarizes the key findings and proposes future research directions. I suggest the following points to be considered in the section:

Comment #9:  Possible explanations for heterogeneity: The discussion could delve deeper into possible reasons for the inconsistent findings across studies, such as variations in participant characteristics (age, comorbidities) or treatment status.

Authors’ reply: We included more explanations for heterogeneity in the discussion section as follows: Additionally, the heterogeneity observed in the measurement of 5-HT levels across studies may stem from various factors beyond differences in laboratory procedures and techniques. Variations in participant characteristics, such as age, the presence of comorbidities, or the accurate exclusion of secondary causes of autism, could contribute to inconsistent findings. For instance, the age range of participants in studies may vary widely, potentially influencing 5-HT levels due to developmental changes. Moreover, comorbid conditions, such as gastrointestinal disorders or mood disorders, which are prevalent in individuals with ASD, could confound the relationship between 5-HT levels and ASD symptoms. Furthermore, inadequate exclusion of secondary causes of autism, such as genetic or metabolic conditions, may result in the inclusion of individuals with heterogeneous etiologies, contributing to the variability in 5-HT levels observed across studies. Additionally, treatment status, including medication usage or dietary interventions, may impact 5-HT levels and introduce further variability in study outcomes. Therefore, it is essential for future research to consider and control for these potential confounding factors to improve the comparability and reliability of findings in the investigation of hyperserotonemia in ASD.

Comment #10: Addressing limitations of serotonin as a biomarker: The authors mention that serotonin levels are not a perfect indicator of brain serotonin function. They could discuss alternative approaches to measuring brain serotonin activity.

Authors’ reply: We included this limitation in the novel  “Limitations and Future Directions” section (see above).

Comment #11: Ethical considerations: Future research on potential uses of serotonin levels in ASD diagnosis or treatment should consider ethical implications.

Authors’ reply: We addressed this issue in the “Limitations and Future Directions” section, as well (see above).

Reviewer #2

Comment #1: The review entitled “A systematic review on Autism and Hyperserotonemia: state of 3 art, limitations and future directions” reported on an interesting topic regarding the presence of hyperserotonemia in subjects with ASD and the correlations between 5-HT levels and ASD-core clinical outcome. The topic is of interest and of actual investigation; and the text and contents are understandable. There are minor concerns:

Authors’ reply: We thank the Reviewer for taking his/her time to review our work and for considering our manuscript of interest.

Comments #2-5: In the abstract line 24 please change in letters the number 1104

Line 40: please make a correction of reference 4

Line 78: In the leterature have benn reported also SSRI use during pregnancy other than such as valproic acid, alcohol, and cocaine. Please specify.

Line 236 : the number 57 should be written in letters.

Authors’ reply: We accepted all the proposed corrections.

Comments #6,  #8-9: Line 324: Platelet Rich Plasma (PRP). Twenty studies analysed 5-HT in PRP. Please add “Regarding platelet rich plasma(PRP): ….etc

Line 339 Platelet Poor Plasma (PPP) or Plasma. Please modify as suggested for line 324

Line 349 Serum. Please modify as suggested for line 324

Authors’ reply: In accordance with another Reviewer’s suggestion, we removed most of the removed most of the 3.3.4 paragraph (Peripheral serotonin concentration) from the main text, including its content as Supplementary Table 3S, in order to improve readability.

Comment #7: It is to note that there is a discrepancy as the authors in line 324 reported that 20 studies analysed 5 HT, instead in the legend of figure 4 only 18 papers have been reported. Please verify.

Authors’ reply: Thank you for pointing this out. The text should have shown “Eighteen studies” instead of 20. We modified the supplementary table 3S accordingly (see previous comment).

Comments #10-12 Line 373: 26 studies. Most often authors reported the number at the start of the sentence. Please edit the numbers in letters: Twenty-six (n=26)

Line 431, 440, 447: Please modify as suggested for line 324

Line 452. In addition, Sacco et al: “claim”, please edit with “claims”

Authors’ reply: We modified the text as suggested.

Comment #13: Line 607. There are ongoing large cohort study groups focusing attention on biomarker in ASD.  In the final discussion please add the reference related to the need to have a large cohort study whose results have not yet  been available Mol Autism. 2017 Jun 23;8:24. doi: 10.1186/s13229-017-0146-8

Authors’ reply: We thank the Reviewer for this suggestion. We included the reference and commented it in the Limitation and Future Direction section, as follows: “Alternative approaches focused on brain serotonin activity (including neuroimaging techniques or the measurement of other serotonin metabolites) have not been considered in the present review, even though they could offer valuable insights and could be found to be more accurate biomarkers for ASD diagnosis or treatment monitoring [24, 67, 105]. The ongoing longitudinal large cohort studies to identify and validate stratification biomarkers for ASD - whose results have not yet been fully available - offer the potential to address some of the limitations observed in individual studies by providing robust data and enhancing the generalizability of findings [184].

Reviewer #3

Thank you for the opportunity to review this paper. This is a timely and important issue to explore that i believ will merit publication. Please kindly consider the following points and revise them.

Authors’ reply: we thank the Reviewer for reviewing our manuscript and for considering it timely and deserving publication, we really appreciate it.

Comment #1: The authors did not present the strengths and limitations of the study.

Authors’ reply: We included a more detailed sub-section about the strengths, limitations, and future directions of this line of studies at the end of the Discussion section (Limitations and Future Directions).

Comment #2: In the whole manuscript: research articles usually do not use the word „our”, „we” and regularly use passive verbs.

Authors’ reply: Thank you for pointing this out. We modified the paper accordingly.

Reviewer 2 Report

Comments and Suggestions for Authors

The review   entitled “A systematic review on Autism and Hyperserotonemia: state of 3 art, limitations and future directionsreported on an interesting topic regarding the presence of hyperserotonemia in subjects with ASD and the correlations between 5-HT levels and ASD-core clinical outcome.

The topic is of interest and of actual investigation; and the text and contents are understandable. There are minor concerns: 

In the abstract line 24 please change in letters the number 1104

Introduction

Line 40: please make a correction of reference 4

Line 78

In the leterature have benn reported also SSRI use during pregnancy other than such as valproic acid, alcohol, and cocaine. Please specify

Line 236 : the number 57 should be written in letters

Line 324: Platelet Rich Plasma (PRP). Twenty studies analysed 5-HT in PRP.

Please add “Regarding platelet rich plasma(PRP): ….etc

It is to note that there is a discrepancy as the authors in line 324 reported that 20 studies analysed 5 HT, instead in the legend of figure 4 only 18 papers have been reported. Please verify.

Line 339 Platelet Poor Plasma (PPP) or Plasma. Please modify as suggested for line 324

Line 349 Serum. Please modify as suggested for line 324

Line 373 : 26 studies

Most often authors reported the number at the start of the sentence. Please edit the numbers in letters :Twenty-six (n=26)

Linea 431, 440, 447: Please modify as suggested for line 324

Line 452

In addition, Sacco et al: “claim”, please edit with “claims”

Line 607

There are ongoing large cohort study groups focusing attention on biomarker in ASD.  In the final discussion please add the reference related to the need to have a large cohort study whose results have not yet  been available Mol Autism. 2017 Jun 23;8:24. doi: 10.1186/s13229-017-0146-8

Comments on the Quality of English Language

The review   entitled “A systematic review on Autism and Hyperserotonemia: state of 3 art, limitations and future directionsreported on an interesting topic regarding the presence of hyperserotonemia in subjects with ASD and the correlations between 5-HT levels and ASD-core clinical outcome.

The topic is of interest and of actual investigation; and the text and contents are understandable. Minor However, there are few concerns that have to be addressed. Minor editing of English language requiredMinor revision has to be considered.

In the abstract line 24 please change in letters the number 1104

Introduction

Line 40: please make a correction of reference 4

Line 78

In the leterature have benn reported also SSRI use during pregnancy other than such as valproic acid, alcohol, and cocaine. Please specify

Line 236 : the number 57 should be written in letters

Line 324: Platelet Rich Plasma (PRP). Twenty studies analysed 5-HT in PRP.

Please add “Regarding platelet rich plasma(PRP): ….etc

It is to note that there is a discrepancy as the authors in line 324 reported that 20 studies analysed 5 HT, instead in the legend of figure 4 only 18 papers have been reported. Please verify.

Line 339 Platelet Poor Plasma (PPP) or Plasma. Please modify as suggested for line 324

Line 349 Serum. Please modify as suggested for line 324

Line 373 : 26 studies

Most often authors reported the number at the start of the sentence. Please edit the numbers in letters :Twenty-six (n=26)

Linea 431, 440, 447: Please modify as suggested for line 324

Line 452

In addition, Sacco et al: “claim”, please edit with “claims”

Line 607

There are ongoing large cohort study groups focusing attention on biomarker in ASD.  In the final discussion please add the reference related to the need to have a large cohort study whose results have not yet  been available Mol Autism. 2017 Jun 23;8:24. doi: 10.1186/s13229-017-0146-8

Author Response

(The authors gave the same response as above.)

Reviewer 3 Report

Comments and Suggestions for Authors

Thank you for the opportunity to review this paper. This is a timely and important issue to explore that i believ will merit publication. Please kindly consider the following points and revise them.

- The authors did not present the strengths and limitations of the study.

- In the whole manuscript: research articles usually do not use the word „our”, „we” and regularly use passive verbs.

Author Response

(The authors gave the same response as above.)

Round 2

Reviewer 1 Report

Comments and Suggestions for Authors

Reviewing a systematic review can be difficult because the review paper requires a high level of expertise in research methodology, critical thinking skills, and meticulous attention to ensure the validity and reliability of the conclusions drawn. However, I believe that the authors have made a concerted effort to address any issues raised and have taken all comments and suggestions into consideration. This updated version is more focused compared to the previous submission. Particularly in the newly added section of "Limitations and Future Directions".